# Examining the New-Member Effect to an Established Community-Based Physical Activity Program for Older Adults in England

**DOI:** 10.3390/ijerph20126161

**Published:** 2023-06-17

**Authors:** Geoff Middleton, Robyn Hambrook, Daniel C. Bishop, Lee Crust, David R. Broom

**Affiliations:** 1School of Sport and Exercise Science, College of Social Sciences, University of Lincoln, Lincoln LN6 7TS, UKleecrust@hotmail.co.uk (L.C.); 2Centre for Sport, Exercise and Life Sciences, Faculty of Health and Life Sciences, Coventry University, Coventry CV1 5FB, UK

**Keywords:** older adults, exercise, physical fitness, elderly, therapeutic exercise

## Abstract

Community-based group physical activity programs promote exercise opportunities for older people. The aim of this study was to examine the short-term, new participant effect after joining Vitality, a community-based group physical activity program available in the East of England for older adults. Two independent groups of participants were assessed before and after an 8 week period: a group recruited from the ‘Vitality’ program (VP) (n 15, age: Age = 69.4 ± 6.4 y), and; a non-intervention control (CON) group (n 14, age: 64.5 ± 5.8 y). Assessment outcomes included basic physical health measures, a fitness test battery, and three psychological scales. The VP group recorded statistically significant improvements on the following outcomes: body mass (VP: −1.39 kg/CON: −0.2 kg), body mass index (VP: −1.5 kg/CON: −0.2 kg), 6 min walk (VP: +42.81 m/CON: −0.45 m), 30 s sit-to-stand (VP: −1.7 s/CON: −0.7 s), the chair sit-and-reach (VP: +3.12 cm/CON: +1.90 cm), and the 30 s arm curl test (VP: + 2 reps/CON: +0.9 reps). No significant differences were found with the other outcomes assessed. New members to the Vitality program achieved several physical and functional benefits without regression on any aspects of physical or psychological health.

## 1. Introduction

Estimates suggest that the global population aged >60 y is set to increase from 841 million in 2013 to more than 2 billion by 2050 [1]. With a growing population of older adults, promoting ‘healthy ageing’ has become a focus for Public Health to reduce the burden of disease and disability including the related healthcare costs [2,3]. It has been illustrated that the older adult population have the highest prevalence of degenerative musculoskeletal conditions such as osteoporosis and sarcopenia [4]. Moreover, mobility can be lost through the ageing process, medical disorders, and disuse of joints which impairs functional independence and capability [5,6]. The prevalence of these varying conditions can cause decline in physical function and increase dependency [7,8]. The age-related deterioration of the body’s physiological and psychological processes, in turn, creates larger demands on public health services [9]. From those admitted to hospital, estimations indicate two-thirds of the people are over 65 years old and this age group accounted for 68% of hospital emergency beds [10]. Multiple mechanisms, including pain, muscle weakness, or decreased proprioception can, singularly or combined, cause an individual to fall [6]. Previous estimates have revealed that one in three adults aged 65–69 years experiences a fall each year [11].

American physical activity guidelines for older adults suggest this population group undertake either 150–300 min of moderate-intensity physical activity (PA) per week, 75–150 min of vigorous-intensity activities per week, or an equivalent combination of the two [12]. Furthermore, muscle strengthening activities involving the major muscle groups twice a week should be completed [12]. These are comparable to the UK Chief Medical Officers [13] guidelines, which also highlight that older adults should build strength and improve balance at least two days a week and minimize time spent being sedentary. Despite revisions of PA guidelines by both the US [12] and the UK governments [13], estimates in the UK report that only 55% of men and 41% of women aged 75 years old and above meet PA recommendations [14]. Previous suggestions have indicated that 60% of older adults are unable to achieve the minimum amount of PA recommended [15]. More specifically, adherence to aerobic and strength training components are markedly low in this age group [16].

Interestingly, when older adults in the community are provided the opportunity to exercise, the risk of falls is reduced [17,18]. Over the last 25 years, PA promotion has been delivered inside and outside the healthcare system, ranging from general advice to exercise prescription schemes where physicians or other healthcare professionals direct specific patients to formal exercise programs, usually within in the community. Community-based group PA programs have been found to have beneficial effects on older adults’ physical functions, such as improved mobility, flexibility, and upper and lower limb function [11,19,20]. Community-based programs allow localized opportunities for group physical activity, which have been found to have positive effects on participants’ subjective sense of wellbeing, health-related quality of life, and mental health [21,22]. Furthermore, mean long-term (≥1 year) adherence rates have been found to be high (70–75% [23,24]) which illustrates the potential for sustained participation when opportunities are provided.

### Public Health Program Background

Operating as a not-for-profit organization located in the East of England, Vitality is subsidized by local government funding and is approved by the National Health Service. Founded in 2009, Vitality has 40 regular classes occurring weekly and located across the county of Lincolnshire, England, UK. The 45–60-min classes are held in various community locations including church halls, sheltered housing, and leisure centers. Each class is supervised by a fitness instructor qualified to work with older and vulnerable adults. Participants are charged a nominal fee of £4 per class after their first session, which is at no cost.

Vitality is designed for adults over 60 years and aims to improve aspects of physical health—functional mobility, coordination, and balance—through physiotherapy-based exercises. These are implemented through dance routines, circuits, and the use of music and equipment, resistance bands, exercise balls, scarves, and pom-poms. A typical Vitality session involves warmup, cardiovascular and muscular endurance exercises (upper and lower body), and cooldown stretches (for a breakdown on specific exercises see Appendix A). Moreover, the program aims to improve psychological and social aspects of health and encourages members to socialize with one another.

There has been great emphasis on building and establishing PA into our societies, communities, and environments [25]. Vitality represents a population-wide intervention which provides the opportunity in the local environment for older adults that join the program to be more active. There is very little known on the effects of implementing programs at broad scale in the community by private providers, which represents a context outside of the traditional controlled scientific trial typically found in experimental research [26,27]. To address the gap in knowledge, the purpose of this study was to examine the new member effect on physiological and psychological health outcomes to the established community-based group PA program. The research team was funded by Vitality to conduct an independent study exploring the effects of this program.

## 2. Materials and Methods

A quasi-experimental approach, specifically a pre-post with a nonequivalent control group design [28], was adopted for this study. This design choice was considered unintrusive because Vitality was an existing Public Health program. This approach is highly regarded for studying effectiveness in real-world settings, achieving balance between internal and external validity [28]. Furthermore, participants could receive exposure to the program without being randomized to keep the equity of the service provision offered by the provider.

Two independent groups of participants were tested before and after an 8 week period: (1) an intervention group recruited from the Vitality program (VP) who were ‘new members’, enrolled before committing to community group exercise classes; and (2) a non-intervention control (CON) group acting as a counterfactual reference for the study (for study protocol see Appendix B (see Figure A1)).

### 2.1. Ethics and Participant Recruitment

Prior to recruiting the participants, the study received institutional ethical approval from an ethics committee at the University of Lincoln (UK, approval number: 20151310) and the work described was carried out in accordance with The Code of Ethics of the World Medical Association (Declaration of Helsinki). The researchers followed the INVOLVE guidelines throughout the study period [29]. Both groups were recruited by a convenience sampling strategy. The VP group recruited fifteen participants (Age = 69.4 ± 6.4 y) via a referral system administered by the VP manager over a four-month period. The manager contacted new participants on enrollment and initially invited new recruits to take part on commencement of their first class. Contact details of the volunteers were provided to a researcher who established contact via telephone or email. For eligibility purposes, participants had to join within four weeks of referral. Participants were excluded by the research team if they could not attend two visits to the study assessment site over the course of 8 weeks and if there were any medical history concerns raised in relation to completing the tests selected by this study. A CON group was recruited involving fourteen participants who were members of the general public (Male n = 6, Female n = 8/Age = 64.5 ± 5.8). Participants were recruited by email and personal enquiry after electronic advertisement within a public sector organization. For the CON group, inclusion criteria stated that volunteers should be between the ages of 55 and 85 years, not currently participating in structured exercise and without injury or medical issues which would impede physical testing. Participants were excluded if the medical history highlighted concerns to the physical and psychological tests in the study. The CHAMPS physical activity questionnaire for older adults [30] was completed by the CON group participants to ensure that they matched the criteria. The questionnaire included over forty items and was formatted so that if participants engaged in a specific activity, they reported the number of times per week (frequency) they completed the activity and also the approximate duration (in hours) of their participation per week [30]. A £30 high street shopping voucher was given to each participant on completion of the test scenario after 8 weeks. All the participants were required to meet screening standards following the American College of Sports Medicine standards and protocols [31].

### 2.2. Procedures

Participants in both the VP and CON groups followed identical assessment procedures before and after the 8 week period. The VP group attended an average of 8.6 ± 2.4 Vitality classes between the two assessment dates. The control participants were asked verbally to refrain from participating in any structured PA during the 8 week period and to keep their activity as ‘normal’. The assessment procedure for each participant entailed the following measurements:

### 2.3. Basic Physical Health Measures

Anthropometric and physical health measurements including stature (cm), body mass (kg), body mass index (BMI) (kg/m^2^), resting heart rate (b/min), systolic and diastolic blood pressure (mmHg). Validated and reliable equipment was used in a university laboratory building, and trained and experienced personnel used standardized operating procedures to ensure the accuracy and reliability of the measures.

### 2.4. Functional Fitness Assessments

A total of six functional fitness assessments were administered: (1) a 6 min walk (6MW), (2) an 8 foot-up-go, (3) a 30 s sit-to-stand, (4) a 30 s arm curl, (5) a back scratch, and (6) a chair sit-and-reach. These were adopted from Rikli and Jones’s [32] ‘Senior Fitness Test Manual’, a battery of test items with high validity and reliability (reliability estimates ranging from 0.81–0.95). These assessments have been used for similar intervention studies involving older adults [11,19] and were conducted with standardized operating procedures [32].

### 2.5. Psychological Scales and Quality of Life

#### 2.5.1. Enjoyment

All the participants completed the physical activity enjoyment scale (PACES) [33]. The questionnaire includes 18 semantic differential items and requires the respondents to select a point along a 7-point continuum between two opposite descriptors relevant to enjoyment of physical activity (e.g., “I enjoy it”… “hate it”, “I like it”… “I dislike it”). The PACES has demonstrated validity and strong internal consistency (a = 0.93), with item-total correlations in the range of r = 0.35–0.89 (Kendzierski & DeCarlo, 1991) [33].

#### 2.5.2. Self-Efficacy

The Self-Efficacy for Exercise (SEE) scale [34] is a 9-item questionnaire that focuses on the self-efficacy expectations for exercise for older adults, specifically the ability to continue to exercise despite certain barriers. The participants were asked about their confidence level on a scale from 0 (not confident) to 10 (very confident) if they would exercise 3 times per week for 20 min during each of the nine hypothetical situations presented. Resnik and Jenkins [34] have reported evidence for validity and an alpha coefficient of 0.92 for internal consistency.

#### 2.5.3. Quality of Life

The SF-36 (36 items) measures quality of life (QOL) across eight domains: (1) physical functioning; (2) role limitations due to physical health; (3) role limitations due to emotional problems; (4) energy/fatigue; (5) emotional wellbeing; (6) social functioning; (7) pain; and (8) general health. Lyons et al. [35] studied the suitability of the Short Form-36 with older adults, illustrating a high degree of internal consistency with Cronbach’s alpha statistic exceeding 0.8 for each parameter as well as construct validity distinguishing between those with and without markers of poorer health [35].

### 2.6. Data Analysis

This study used an average overall summarized questionnaire score as a main outcome variable for each instrument. Using a statistical software package (IBM SPSS statistics, V26), a 2 × 2 mixed-analysis of variance (ANOVA) with repeated measures was performed to calculate the statistical difference between all measured outcomes (dependent variables) at the data collection time points (baseline and 8 weeks) between groups (VP and CON) to assess whether an interaction existed (Time × Group). Prior to any inferential statistical analysis, all the study outcome data was checked for outliers and normality (via a Shapiro–Wilk test; *p* ≥ 0.05). Paired *t*-tests were deployed to assess pre–post differences between variables for each group separately. Any significant differences detected have been shown with an accepted *p* value of <0.05. Effect sizes were calculated using Cohen’s *d* and should be interpreted as small, medium, or large effects with cutoffs as 0.2, 0.5, and 0.8, respectively. Data are presented as means (M) and standard deviations (SD) unless otherwise stated.

## 3. Results

### 3.1. Basic Physical Health Measurements

There was a main effect of time on the body mass (*F*(1, 27) = 10.668, *p* = 0.03, *η_p_^2^* = 0.283) and BMI assessments (*F*(1, 27) = 9.248, *p* = 0.005, *η_p_^2^* = 0.255). No other main effects were detected (all *p* ≥ 0.05) with other physical health measures. The ANOVA revealed a significant interaction between time and group for body mass (*F*(1, 27) = 6.667, *p* = 0.016, *η_p_^2^* = 0.198). No other interactions were found with other health measures (all *p* ≥ 0.05). Paired *t*-tests found that both body mass (*t*(14) = 3.374, *p* = 0.005) and BMI (*t*(14) = 2.824, *p* = 0.014) were significantly reduced from baseline to 8 weeks in the VP group. No other significant difference was detected. In contrast, the CON group remained relatively the same over time with screening measures with marginal changes in mean performance before and after 8 weeks (all *p* ≥ 0.05). The results can be viewed in Table 1.

### 3.2. Functional Fitness Measurements

The functional fitness results are presented in Table 2. There was a significant main effect of time on the 6MW (*F*(1, 27) = 7.705, *p* = 0.010, *η_p_^2^* = 0.222), 30 s sit-to-stand (*F*(1, 27) = 12.761, *p* = 0.001, *η_p_^2^* = 0.321), and 30 s arm curl, (*F*(1, 27) = 14.036, *p* = 0.001, *η_p_^2^* = 0.342). No further main effects were found (all *p* ≥ 0.05) with other fitness measurements. A significant interaction between time and group was detected for the 6MW test (*F*(1, 27) = 12.072, *p* = 0.002, *η_p_^2^*2 = 0.309) and the chair sit-and-reach (*F*(1, 27) = 4.544, *p* = 0.042, *η_p_^2^*2 = 0.144). No other interactions were found with other fitness assessments (all *p* ≥ 0.05). Paired *t*-test results showed the VP group to improve significantly over time with the 6MW (*t*(14) = −5.025, *p* = 0.000), 30 s chair sit-to-stand, (*t*(14) = −3.833, *p* = 0.002), 30-s arm-curl, (t(14) = −1.100, *p* = 0.003) and the chair sit-and-reach (*t*(14) = −2.338, *p* = 0.035) tests. In comparison, no improvements over time were detected with the CON group for the same outcomes (all *p* ≥ 0.05).

### 3.3. Psychological Scales and QOL

The ANOVA did not detect any main effect of time with SEE, PACES, or QOL outcomes (all *p* ≥ 0.05). Furthermore, no significant interactions were found between time and group (all *p* ≥ 0.05). Paired *t*-tests revealed that overall, both the VP and the CON group showed marginal mean changes in these assessments over 8 weeks without any significance detected (all *p* ≥ 0.05). Table 3 highlights the findings.

## 4. Discussion

The aim of this study was to examine the efficacy of Vitality for new members to the established community-based group exercise program. Key findings suggest that Vitality has beneficial effects on body mass, 6 min waking distance, and lower-body flexibility for older adults. In addition, whilst there were no significant improvements, there were no significant regressions in QOL, self-efficacy, exercise enjoyment, agility, muscular strength, or upper-body flexibility, meaning that participation in Vitality did not lead to any negative outcomes. Whilst no significant differences were found between the VP and CON participants with the 8UG, 30 s chair sit-to-stand and arm curl tests, and with PACEs and SEE questionnaires, effect sizes were consistently larger in the VP group.

The VP group significantly improved performance in the 6MW (9%) from baseline to 8 weeks compared to the CON group. This is indicative of improved aerobic capacity [36], which is fundamental to undertaking activities of daily living such as walking to the shops without experiencing excessive fatigue. Greater improvements to 6MW have been observed [37] with older individuals who have suffered from chronic stroke (19.6%; 10% greater than the present study); however, the population achieved a lower baseline distance score, likely due to their condition. Pang et al. [37] suggest that the most effective method to improve aerobic endurance is standing rather than chair-based exercise. Participants completed all three types of Vitality class: standing, seated, and a combination of both; therefore, it is difficult to identify which type of class is most beneficial for improving 6MW, and this warrants further investigation. Training the aerobic energy system decreases the stress on the body for any given work rate, consequently improving an individual’s ability to walk longer distances, shop, and engage in sightseeing [38]. It has been reported that aerobic capacity can be lost at a rate of 10% per decade, so maintaining an active lifestyle can lessen the rate of decline and improve the chances of living independently free from disease [39].

Although improvements were noted in the VP group with the 30 s chair sit-to-stand (13%), the 8UG (7%), and the 30 s arm curl (12%) tests from baseline to 8 weeks, these results were not significant in comparison to the CON, albeit large effect sizes are reported. The physiological effects of resistance training are the result of numerous factors including genetics, health status, and training dose (frequency/intensity/time). Previous research demonstrates that 2–3 sessions of resistance training per week at an intensity of 65–85% of 1 repetition maximum is most successful at improving muscle strength in older adults [40]. Although the VP group illustrated improvements in our study, greater increases with arm curl and chair sit-to-stand tests have been reported in other community-based exercise programs using traditional resistance training equipment [11]. Despite relatively limited resources, the VP group did illustrate improvements, which is encouraging for community-based programs which have limited resources and access to gym settings or similar environments.

Interestingly, back scratch results (upper body flexibility) from baseline to 8 weeks showed no significant improvement in either group, whereas the results of the chair sit-and-reach (lower body and trunk flexibility) showed the VP group significantly improved. Range of motion in the shoulder (abduction and adduction) and hip (flexion and extension) decline with age, and flexibility exercises which challenge these joints have been advocated [41]. Despite this, it may be challenging for older adults to improve upper body flexibility in a short time frame. Only small improvements (3.5%) were found in an intervention group after 6 months of regular training with the use of elastic bands [42]. Upper body movements are essential for older adults to maintain the activities of daily living (ADL) such as washing hair and stretching for a high object; despite the difficulty of improving this aspect, it shouldn’t be neglected as part of a well-balanced exercise program. Therefore, it was positive that the VP group managed to improve upper-body flexibility, and this is likely to be attributable to a variety of exercises in the program encouraging upper-body movement (see Appendix A), even if the patient is seated or standing.

Whilst this study has strengths, including the novel collection of data from a real-life Public Health intervention based in a community setting, it has limitations which should be addressed. The study sample was relatively small, due to limited funding and time. A larger sample of participants would have increased the statistical power associated with our findings, but we have presented effect sizes to allow the user to interpret the findings, especially when significance was not achieved by more traditional inferential statistics. In addition, a firmer match between participant characteristics in both groups may have been achieved (i.e., reduce the age difference between groups) with a larger sample. Recruiting from a ‘live’ Public Health program over the course of four months restricted the participant pool available. Furthermore, results of the CON group indicated a slightly ‘fitter’ participant group in many of the assessments (e.g., blood pressure, 6MW test). This may be explained by the slightly younger age of this group compared to the VP group.

Vitality continues to deliver PA to the community via a network of instructors at several venues in one geographical region in the UK. Inevitably, at a community setting level, Vitality class provision will vary for participant ability levels and other contextual factors (availability of equipment, facility space, etc.). The research team involved in the evaluative work did not have influence over the content of the sessions delivered, as this was an existing community-based Public Health initiative. A worthwhile aspect for the Vitality program would be to consider evaluating the exercise progression and fidelity of new but also existing members, which has occurred in other exercise interventions for older adults [43]. It should be acknowledged that whilst participants in both the VP and CON groups were instructed not to participate in any additional daily physical activity, this was not monitored during the study period. Similarly, although our findings revealed a significant improvement in body mass in the VP participants, little is known on the body composition changes which may have occurred in this time frame with older adults. Nevertheless, it is important to acknowledge that the associated benefits of physical activity are more important than body mass loss per se, including favorable changes to blood lipids, improved blood pressure [44], and allowing the opportunity for social connection [20].

## 5. Conclusions

This study provides evidence that participation in Vitality, a community-based group PA program, can positively affect body mass and physical function including aerobic fitness and lower-body flexibly in older adults over an 8 week period. Although little improvement was made in psychological health in this study, there was no sign of regression with the participants on these outcomes. In this light, Vitality is recognized as a promising health promotion program for improving physical health over a short time period within the community for the older adults that join. The sustainability of the effects remain to be elucidated.

## Figures and Tables

**Table 1 ijerph-20-06161-t001:** Participant characteristics and physical health screening measurements; group comparison.

Assessment/Group	Baseline	After 8 Weeks	Difference (+/−)	Effect Size (Cohen’s *d*)
Stature (cm)				
Vitality:	159.7 ± 3.9	159.3 ± 4.0	−0.4	
Control:	166.3 ± 11.1	166.5 ± 11.2	+0.2	
Body mass (kg)				
Vitality:	73.3 ± 8.6	71.8 ± 8.5 *	−1.5 **	0.17
Control:	76.8 ± 20.6	76.6 ± 20.4	−0.2	0.01
Body Mass Index (kg/m^2^)				
Vitality:	28.8 ± 3.4	28.3 ± 3.3 *	−0.5	0.15
Control:	27.5 ± 6.0	27.4 ± 5.9	−0.1	0.01
Resting Heart rate (b/min)				
Vitality:	74.8 ± 15.4	76.5 ± 11.8	+1.7	0.01
Control:	76.4 ± 9.7	75.5 ± 12.0	−0.9	0.08
Systolic pressure (mmHg)				
Vitality:	145.8 ± 17.2	145.6 ± 21.3	−0.2	0.09
Control:	138.1 ± 15.8	134.4 ± 14.1	−3.7	0.24
Diastolic pressure (mmHg)				
Vitality:	79.5 ± 10.4	78.7 ± 10.3	−0.8	0.07
Control:	80.0 ± 11.2	76.9 ± 6.6	−3.1	0.33

* denotes a statistical significant difference (*p* ≤ 0.05) between baseline and 8 weeks. ** denotes a significant interaction between time and group (*p* ≤ 0.05).

**Table 2 ijerph-20-06161-t002:** Functional fitness assessment results; group comparison.

Assessment/Group	Baseline	After 8 Weeks	Difference (+/−)	Effect Size (Cohen’s *d*)
6 min walk (m)				
Vitality:	430.5 ± 38.1	473.4 ± 37.0 *	+42.8 **	1.14
Control:	521.8 ± 73.3	517.0 ± 69.8	−4.8	0.06
30 s chair stand (repetitions)				
Vitality:	11.20 ± 1.9	12.9 ± 2.1 *	+1.7	0.84
Control:	13.9 ± 2.8	14.6 ± 2.8	+0.7	0.25
8 foot up and go (s):				
Vitality:	6.5 ± 1.0	6.00 ± 1.1	−0.5	0.47
Control:	6.1 ± 1.0	6.2 ± 1.2	+0.1	0.09
30 s arm curl (repetitions)				
Vitality:	14.1 ± 2.7	16.1 ± 2.1 *	+2.0	0.82
Control:	17.1 ± 3.3	18.0 ± 3.7	+0.9	0.25
Back scratch (cm)				
Vitality:	−7.2 ± 10.1	−6.0 ± 10.0	+1.2	0.11
Control:	−5.9 ± 8.6	−7.0 ± 8.6	−1.1	0.12
Chair sit and reach (cm)				
Vitality:	−2.3 ± 11.2	0.6 ± 10.00 *	+2.9 **	0.27
Control:	−4.1 ± 10.0	−4.9 ± 9.5	−0.8	0.08

* denotes a statistical significant difference (*p ≤* 0.05) between baseline and 8 weeks. ** denotes a significant interaction between time and group (*p ≤* 0.05).

**Table 3 ijerph-20-06161-t003:** Psychological scale and quality of life assessments; group comparison.

Assessment/Group	Baseline	After 8 Weeks	Difference (+/−)	Effect Size (Cohen’s *d*)
Self-efficacy (SEE) questionnaire (9-item score)				
Vitality:	6.38 ± 1.45	6.88 ± 2.02	+0.50	0.28
Control:	6.84 ± 1.74	6.67 ± 2.07	−0.17	0.08
Enjoyment (PACES) questionnaire (18-item score)				
Vitality:	5.80 ± 0.89	6.13 ± 1.0	+0.33	0.35
Control:	5.77 ± 1.20	5.95 ± 0.70	+0.18	0.18
SF-36 (QOL) questionnaire (36-item score)				
Vitality:	75.2 ± 14.6	78.7 ± 14.3	+3.5	0.24
Control:	80.9 ± 7.6	82.7 ± 7.3	+1.8	0.24

## Data Availability

Data are unavailable due to contractual obligations with the provider/funder of this study.

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
