# Peer review of "Examining the New-Member Effect to an Established Community-Based Physical Activity Program for Older Adults in England"

_ijerph, 2023, doi:10.3390/ijerph20126161_

Round 1

Reviewer 1 Report

MDPI: international journal of environmental research and public health

I appreciate the editor(s) for providing me the opportunity to review a valuable manuscript entitle” Examining the New Member Effect to an Established Community-Based Physical Activity Program for Older Adults in England” :The title is interesting however it needs some clarification before any decision making.

Regarding the manuscript, the following modifications are recommended:

Modification of writing must be considered in all parts of the manuscript.

Title: the main aim is not clear

Abstract:

- again the aims have not been clarified

- writing of ( n 15, age: Age = 69.4 ± 6.4 y) or  (n 14, age: 64.5 ± 5.8 y)  for both groups must be modified and uniform .

- line 15: "the use of" needs to be deleted.

-line 16: unit of mi must be modified.

Introduction

-In the introduction, it must be stated why vitality is different physiologically from similar programs specifically on the study variables.

-Introduction needs the resynchronization of paragraphs in order to reach the aim of the study.

-line 33-34 need a little clarification.

Methods

The first sentences which describe the training method must be moved to a separate tile and must be completed after the study design and describing the participants.

-In lines 114 and 120 it is recommended to write age ranges in the method and mean ± SD move to results. (do not forget units).

-Inclusion and exclusion criteria must be clarified for each group in detail.

-for every research tool mentioning validated is not enough, in all measurements the tools must be introduced completely and if available the devise validity must be mentioned by reference.

- the measurement of physical health screening needs clarification. I also think this title is not suitable.

Discussion

-the changes can be interpreted a bit more according to the components of the training program.

Conclusion

-          It must be obtained from the study findings, some parts can be excluded.

English writing needs minor revisions in all parts.

Author Response

Reviewer 1 Feedback

Author adjustments and comments

Title: the main aim is not clear

The title begins with the word ‘Examining’ as this corresponds exactly to the role of the research team for this project and was the aim of the project; to examine the new member effect of a ‘live’ Public Health programme which was delivered by a public sector organisation (not the research team themselves).  The research team was requested to examine this programme. See additional sentence on lines on lines 94 to 95 which highlights this.

Abstract: again the aims have not been clarified

Amended.

The word purpose has been replaced with ‘aim’ on line 10. 

Abstract: writing of ( n 15, age: Age = 69.4 ± 6.4 y) or  (n 14, age: 64.5 ± 5.8 y)  for both groups must be modified and uniform .

Amended.

We have proofed to ensure the consistency in the presentation of all our data. Despite multiple authors checking this please alert us to any instances that are not to your satisfaction. Yes, there was a slight difference. We have added lines 308-314 to the discussion regarding the limitations on this aspect. 

Abstract: line 15: "the use of" needs to be deleted.

Amended.

Thank you for pointing this out, this has been changed accordingly.

Abstract: line 16: unit of mi must be modified.

Not amended.

The 6-minute walk test is constructed using metres as a marker for the test area and thus using metres covered (and not miles; mi) for distance covered is the primary and direct measure for this test. Older participants who complete this test at the ages we recruited in this study, would not achieve more than 1 mile during each test.  

Introduction: In the introduction, it must be stated why vitality is different physiologically from similar programs specifically on the study variables.

Amended.

The section on the Public Health background (69 – 95) outlines the specific context of this as a community-based programme. Vitality is therefore using the science that already exists which informs practice; that low-moderate intensity exercise is beneficial to older adults who should have access to this in their communities. The research team was requested to examine this programme. See additional sentence on lines on lines 94 to 95 which highlights this. Furthermore, information in Table 1 outlines the specific exercises and coverage. Readers are therefore informed of the exercises and can use this information if necessary to compare interventions/programmes themselves hopefully citing this article in the future.

Introduction: Introduction needs the resynchronization of paragraphs in order to reach the aim of the study.

Amended.

Thank you for this comment, we have adjusted the structure of the Introduction and have acted on this effectively by swapping the initial paragraph for the second for a better flow to the document. The reference numbering system has also been adjusted according to this change.

Introduction: line 33-34 need a little clarification

Amended.

Due to changes outline in the previous comment, these lines are now 49 - 52 and this corresponds to the following sentences which reads:  “Despite revisions on PA guidelines, by both American [12] and UK governments [13], estimates in the UK report that only 55% of men and 41`% of women aged 75 years old and above meet PA recommendations [14]. Previous suggestions have indicated that 60% of older adults are unable to achieve the minimum amount of PA recommended [15].”

We hope the you can provide the clarification on what is demanded here, please do expand on your expectations on this item. 

Methods: The first sentences which describe the training method must be moved to a separate tile and must be completed after the study design and describing the participants.

Amended.

Vitality is a ‘live’ Public Health programme which the research team examined, this required explanation before the materials and methods sections to provide the context relevant for the study’s design and instruments which are then applied to this situation. However, further clarity on this is now highlighted in lines 93-94 which a statement which highlighted the research team’s role.

Methods: In lines 114 and 120 it is recommended to write age ranges in the method and mean ± SD move to results. (do not forget units).

Not amended.

We do not feel this is a useful action and it is appropriate to report mean and SD of age in the methods section before the results as these highlight the exact sample in a pure and primary descriptive sense.  

Methods: Inclusion and exclusion criteria must be clarified for each group in detail.

Amended.

Thank you for this comment. The Ethics and participant recruitment (page 3) has been expanded in two areas with additional details present. Please see lines 121 to 124 (Vitality group) and 129 to 131 (Con group).

Methods: for every research tool mentioning validated is not enough, in all measurements the tools must be introduced completely and if available the devise validity must be mentioned by reference.

Amended.

Thank you for this comment. Validity has been mentioned in the physical screening testing section. For the functional fitness assessments and the psychological scales, we have entered additional clarity on the validity concerns raised. It is not our intention to conduct a validity or reliability study with these items but do importantly cover these adequately despite being minor aspects of this study.  Please do expand on your expectation on this item if necessary.

Methods: the measurement of physical health screening needs clarification. I also think this title is not suitable.

Amended.

This section title has now been changed and is now titled; Basic physical health measures.

Discussion: the changes can be interpreted a bit more according to the components of the training program.

Amended.

Thank you for this comment. We have added a sentence to cover this on the upper body flexibility aspect of the program improvement (see lines 299 to 301).   

We would also like to point out that we do cover this aspect in lines 267 to 270 and acknowledge that it is difficult to really distinguish the causal factors from the components of the community-based programme for any improvements and that this requires further investigation.

Conclusion: It must be obtained from the study findings, some parts can be excluded.

Amended.

Thank you for this comment and we have been more concise. The conclusion has been shortened from ten lines to eight with two sentences deleted according to your point on relating to the study findings only. 

Reviewer 2 Report

Dear Editors and authors,

Thank you for inviting me to review this manuscript, and congratulations for your work. However, there are some opportunities for improvement, as it is outlined below.

Introduction

Line 50 and line 53: The authors should write "years" instead of "y".

Materials and Methods

Line 188: Did data followed a normal distribution?. Which was the resulting p-value of the normality test? And which normality test was used?

Results

Line 235: "Tale 3" is a writting mistake, it should say "Table 3". Please fix it.

Discussion 

Line 254: The authors should say "to understand" instead of "to understanding".

English is fine, minor spelling mistakes have been reported so authors con fix them.

Author Response

Reviewer 2 Feedback

Author adjustments and comments

Introduction: Line 50 and line 53: The authors should write "years" instead of "y".

Amended.

Thank you for pointing this out, this has now been changed throughout the document.

Materials and Methods: Line 188: Did data followed a normal distribution?. Which was the resulting p-value of the normality test? And which normality test was used?

Amended.

Thank you for the comment. The answers to these queries have now been provided. Due to changes to the paper, this is now lines 199 and 200.

Results: Line 235: "Tale 3" is a writing mistake, it should say "Table 3". Please fix it.

Amended.

Thank you for pointing out this mistake, this has now been changed to ‘Table 3’.

Discussion: Line 254: The authors should say "to understand" instead of "to understanding".

Not amended.

The word in line 262 (after changes) is ‘undertaking’, not understanding. This has not been changed as we believe it to be grammatically correct.

Reviewer 3 Report

Thank you very much for inviting me to revier the paper entitled: Examining the New Member Effect to an Established Community_Based Physical Activity Program for Older Adults in England.

General comments: A well-written and scientifically interesting manuscript that addresses an important area of medical need. The purpose of this study was to examine the short-term, new participant effect after joining “Vitality”; a community-based group physical activity program available in East of England for older adults.

Please see my specific comments below for more details,

This manuscript consists of a non-structured abstract with 3 keywords, 5 sections (introduction with 1 subsection, materials & methods with 6 subsections, results with 3 subsections, discussion with limitations of the study, and conclusions) on 13 pages of single-spaced text with embedded tables (3), appendix (2: A: Example exercises from a typical Vitality class & B: the study protocol). There are 44 references.

Specific comments:

  1. The authors have used 3 keywords, they should use some more such us “elderly, therapeutic exercise…”.
  2. Introduction: I consider the subsection 1.1Public Health Program Bakcground” should be in section 2. Materials and methods.
  3. Materials and methods: The methodology is rigorous. A quasi-experimental approach, specifically a pre-post with a non-equivalent control group design. It is posible to add pictures of exercises program in the appendix A: Example exercises from a typical Vitality class?

It is possible to add the code of the Ethics Committee of School of Sport and Exercise Science, University of Lincoln?

As for the inclusion criteria, no type of pathology has been taken into account?

One question, why have participants been rewarded with £30 vouchers?

The scales and questionnaires used in absolutely fine measurements (6MW; 8 foot-up-go; 30 second sit-to stnd… PACES, SEE, SF-36 QOL.

4.      Results: The authors conclude that Vitality is recognized as a promising health promotion program to improve physical health for a short period of time within the community for older adults who join, but they have not conducted the research comparing a control group with other types of exercises. Scientifically it is proven that exercise improves the quality of life in older people, why this program and not another?

5.      The bibliography is current and encompasses the most recent scientific advances for research.

Thanks again for the invitation.

Author Response

Reviewer 3 feedback

Authors adjustments and comments

Keywords: The authors have used 3 keywords, they should use some more such us “elderly, therapeutic exercise…”.

Amended.

Thank you for this comment, we have now included these as keywords.

Introduction: I consider the subsection 1.1Public Health Program Background” should be in section 2. Materials and methods.

Not amended.

This description is useful before materials and methods section because this sets the context for the study and the background of the Public Health programme (i.e. what the following procedures and instruments are applied to). After publishing several articles in various peer reviewed journals on similar projects, we have never placed a description of a community-based programme in the recognisable ‘methods’ section.

Materials and methods: The methodology is rigorous. A quasi-experimental approach, specifically a pre-post with a non-equivalent control group design. It is posible to add pictures of exercises program in the appendix A: Example exercises from a typical Vitality class?

Not amended as not possible.

Pictures were not authorised for use in this manner / during ethical clearance. We have provided a breakdown in text of the typical exercises expected in the Vitality program in Appendix A. We will however consider this in future research.

It is possible to add the code of the Ethics Committee of School of Sport and Exercise Science, University of Lincoln?

Amended.

Number is now provided in line 112.

As for the inclusion criteria, no type of pathology has been taken into account?

Amended.

Thank you for this comment. The Ethics and participant recruitment (page 3) has been expanded in two areas with additional details present.  Please see lines 121 to 124 (Vitality group) and 129 to 131 (Con group).

One question, why have participants been rewarded with £30 vouchers?

Answer:

Participants had to travel and locate themselves at University facility site for the assessments and we used the funding received to attract participants across a large county in England. In some cases, participants travelled 10 miles to make the assessment appointments, so this was our way of compensating them for their time and expense.

The scales and questionnaires used in absolutely fine measurements (6MW; 8 foot-up-go; 30 second sit-to stnd… PACES, SEE, SF-36 QOL.

Thank you for this comment we always seek to use optimal tools and methods for our research.

Results: The authors conclude that Vitality is recognized as a promising health promotion program to improve physical health for a short period of time within the community for older adults who join, but they have not conducted the research comparing a control group with other types of exercises. Scientifically it is proven that exercise improves the quality of life in older people, why this program and not another?

Not amended.

The findings highlight that quality of life scores did not regress and stayed (pre to post) and between groups. We did not state that QOL is better or improves because of the Vitality programme.  

The bibliography is current and encompasses the most recent scientific advances for research.

Thank you for this comment.

Reviewer 4 Report

Congratulations on the search. Below are some suggestions. 

Introduction

Between lines 51 and 53 it is important to add other systems that participate in postural balance

Materials and Methods

Was a convenience sample? 

There was difference between age of groups? 

Results

Some results show that the control group was better than the VP group before the intervention (6mw, 30 second chair stand). It is necessary to show this comparison.

Conclusion

"These fitness components are essential for the completion of various ADLs; stair-climbing, shopping, dressing, and bathing. The importance of maintaining functional fitness is crucial for maintaining an independent lifestyle for older adults." This is not conclusion.

References

75% of references have more than five years of publication. It is necessary to update. 

Author Response

Reviewer 4 feedback

Authors adjustments and comments

Introduction: Between lines 51 and 53 it is important to add other systems that participate in postural balance.

We would be grateful for clarification on what you mean by ‘systems’ so that we can address the reviewer’s point. There is evidence to suggest that the exercises undertaken as part of the Vitality programme will improve postural balance.

Materials and Methods: Was a convenience sample? There was difference between age of groups?

Amended.

Thank you for the comment, lines 115-116 now mentions convenience sampling.

Yes, there was a slight difference. We have added lines 308-314 in the limitations on this aspect. 

Results: Some results show that the control group was better than the VP group before the intervention (6mw, 30 second chair stand). It is necessary to show this comparison.

Amended.

Yes, there was a slight difference. We have added lines 308-314 in the limitations on this aspect, but we feel the results should remain as provided, due to the statistical analysis which was conducted. 

Conclusion: These fitness components are essential for the completion of various ADLs; stair-climbing, shopping, dressing, and bathing. The importance of maintaining functional fitness is crucial for maintaining an independent lifestyle for older adults." This is not conclusion.

Amended.

This has now been removed from this section and was also commented on by other reviewers.

References: 75% of references have more than five years of publication. It is necessary to update.

Not amended.

We have referenced relevant studies and another reviewer has commented positively on the inclusion of specific references and research on the matter.

Round 2

Reviewer 1 Report

can be accepted with minor writing revision

can be accepted with minor writing revision

Reviewer 3 Report

The paper was significantly improved from the last version. Thank you for making changes to the manuscript.